# The AP-1 Transcription Factor Fosl-2 Regulates Autophagy in Cardiac Fibroblasts during Myocardial Fibrogenesis

**DOI:** 10.3390/ijms22041861

**Published:** 2021-02-13

**Authors:** Jemima Seidenberg, Mara Stellato, Amela Hukara, Burkhard Ludewig, Karin Klingel, Oliver Distler, Przemysław Błyszczuk, Gabriela Kania

**Affiliations:** 1Center of Experimental Rheumatology, Department of Rheumatology, University Hospital Zurich, University of Zurich, 8952 Schlieren, Switzerland; jemimaseidenberg@gmail.com (J.S.); Mara.Stellato@usz.ch (M.S.); Amela.Hukara@usz.ch (A.H.); Oliver.Distler@usz.ch (O.D.); or przemyslaw.blyszczuk@uj.edu.pl (P.B.); 2Institute of Immunobiology, Cantonal Hospital St. Gallen, 9007 St. Gallen, Switzerland; burkhard.ludewig@kssg.ch; 3Institute of Experimental Immunology, University of Zurich, 8057 Zurich, Switzerland; 4Cardiopathology, Institute for Pathology and Neuropathology, University Hospital Tubingen, D-72076 Tubingen, Germany; Karin.Klingel@med.uni-tuebingen.de; 5Department of Clinical Immunology, Jagiellonian University Medical College, 30-663 Krakow, Poland

**Keywords:** autophagy, cardiac fibrosis, cardiac fibroblasts, Fos-related antigen 2, cardiac hypertrophy

## Abstract

Background: Pathological activation of cardiac fibroblasts is a key step in development and progression of cardiac fibrosis and heart failure. This process has been associated with enhanced autophagocytosis, but molecular mechanisms remain largely unknown. Methods and Results: Immunohistochemical analysis of endomyocardial biopsies showed increased activation of autophagy in fibrotic hearts of patients with inflammatory cardiomyopathy. In vitro experiments using mouse and human cardiac fibroblasts confirmed that blockade of autophagy with Bafilomycin A1 inhibited fibroblast-to-myofibroblast transition induced by transforming growth factor (TGF)-β. Next, we observed that cardiac fibroblasts obtained from mice overexpressing transcription factor Fos-related antigen 2 (Fosl-2tg) expressed elevated protein levels of autophagy markers: the lipid modified form of microtubule-associated protein 1A/1B-light chain 3B (LC3BII), Beclin-1 and autophagy related 5 (Atg5). In complementary experiments, silencing of Fosl-2 with antisense GapmeR oligonucleotides suppressed production of type I collagen, myofibroblast marker alpha smooth muscle actin and autophagy marker Beclin-1 in cardiac fibroblasts. On the other hand, silencing of either LC3B or Beclin-1 reduced Fosl-2 levels in TGF-β-activated, but not in unstimulated cells. Using a cardiac hypertrophy model induced by continuous infusion of angiotensin II with osmotic minipumps, we confirmed that mice lacking either Fosl-2 (Ccl19CreFosl2flox/flox) or Atg5 (Ccl19CreAtg5flox/flox) in stromal cells were protected from cardiac fibrosis. Conclusion: Our findings demonstrate that Fosl-2 regulates autophagocytosis and the TGF-β-Fosl-2-autophagy axis controls differentiation of cardiac fibroblasts. These data provide a new insight for the development of pharmaceutical targets in cardiac fibrosis.

## 1. Introduction

Cardiac fibrosis is characterized by the excessive deposition of type I collagen and other extracellular matrix proteins in the cardiac tissue. In non-inflammatory heart disease, this process is mediated mainly by activated cardiac fibroblasts and pathological myofibroblasts [1]. Fibrotic changes in the myocardium cause stiffening of the ventricles that may lead to development of cardiomyopathies and impairment in systolic and diastolic heart functions [2]. It has been hypothesized that autophagy may play an important role in cardiac fibrogenesis.

Autophagy, also called autophagocytosis, refers to processes of controlled cellular self-degradation and recycling in response to nutrient deprivation and other cellular stresses. Activation of the autophagocytosis is essential not only for cell survival but plays an important role also in cell homeostasis and during cell differentiation [3]. Autophagy is an evolutionarily conserved process, in which the double membrane vesicle (phagophore) forms autophagosome with cargo proteins that in the next step fuses with a lysosome for the digestion of the content. This process is tightly regulated by specific effector molecules, including a number of autophagy-related (Atg) proteins, Beclin-1 and the cytosolic microtubule-associated protein light-chain 3 beta (LC3B), that represent the most common markers for autophagy monitoring. The family of Atg proteins plays a crucial role in formation of the phagophore, whereas Beclin-1 is an adaptor protein that enables the recruitment of other autophagy proteins involved in the nucleation of the autophagosome. In case of LC3B, a cleaved form of the cytosolic LC3B (LC3B I) is conjugated to phosphatidylethanolamine forming LC3B II that is attached on the surface of newly generated autophagosomes and promotes its maturation [4].

In homeostatic condition, autophagy plays a cardioprotective role by mediating organelle turnover, recycling aged or damaged cytoplasmic components, thereby protecting from aging and development of cardiovascular disease [5]. In particular, defective autophagy leads to cardiac hypertrophy associated with cardiac fibrosis and contractile dysfunction [6,7,8]. Unlike in homeostasis, the role of autophagocytosis in fibrotic heart disease is less clear. In hypertrophic heart, autophagic processes are activated and studies on Beclin-1 suggested that autophagy enhanced tissue remodeling and disease progression [9,10]. In line with these results, treatment with autophagy inhibitor chloroquine ameliorated diastolic function and cardiac fibrosis in diabetic mice [11]. On the other hand, genetic deficiency of autophagy effector protein Atg5 led to increased hypertrophy and cardiac fibrosis [8,12]. Similarly, blunting autophagy accelerated cardiac fibrosis and ventricular dysfunction in a mouse model of desmin-related cardiomyopathy [13].

Involvement of autophagy to cellular and molecular processes is cell type-specific. A growing body of evidence suggests that the autophagy plays a critical role in cardiomyocytes [14]. Unlike for cardiomyocytes, much less is known about the role of autophagocytosis in cardiac fibroblasts. In vitro experiments showed enhanced autophagy in activated cardiac, but also dermal fibroblasts undergoing fibroblast-to-myofibroblast transition [15,16]. Inhibition of autophagocytosis by treatment with bafilomycin A1 (BafA1) and chloroquine reduced formation of myofibroblast and cell mobility [17]. Fibroblast-to-myofibroblast transition can be induced by various molecular triggers, however transforming growth factor (TGF)-β represents a master profibrotic cytokine that most effectively activates myofibroblastic fate of cardiac fibroblasts [18,19]. Of note, TGF-β signaling has been suggested to promote LC3B I to LC3B II conversion and autophagosome formation through its canonical and non-canonical pathways [18,20]. Profibrotic TGF-β signaling is well-known to interplay with other profibrotic pathways [21]. Fos-like 2 (Fosl-2, known also as Fra-2) represents a transcription factor of the activator protein (AP)-1 family that like TGF-β has been associated with fibrotic processes. Mice overexpressing Fosl-2 (Fosl-2^tg^) develop spontaneous autoimmunity and multiorgan fibrosis including skin, lung and the heart [22,23,24]. So far, regulation of autophagocytosis by Fosl-2 remains unknown.

## 2. Results

### 2.1. Autophagy Enhances Myofibroblast Activation of Cardiac Fibroblasts

To evaluate the relevance of autophagy in fibrotic heart disorders, we analyzed expression of autophagy-related proteins LC3B, Beclin-1 and ATG5 in endomyocardial biopsies obtained from iDCM patients who developed fibrotic phenotype in comparison to patients with healed myocarditis (no fibrosis). Myocardial tissue of iDCM patients was characterized by increased fibrosis and elevated levels of autophagy markers: ATG5, Beclin-1 and LC3B (Figure 1). Such association might suggest potential involvement of autophagy in cardiac fibrosis.

Serum deprivation is a common method to activate autophagy in vitro. In the first set of experiments, we analyzed how cardiac fibroblasts upregulate autophagy-associated proteins in response to a switch from 10% to 1% serum in cell culture medium. As expected, we observed upregulation of LC3B I and LC3B II proteins (Figure 2A,B and Appendix A). It should be noted that the LC3B II form is mostly associated with formation of autophagosomes. Upon exposure to cell culture medium containing 1% serum, mouse and human cardiac fibroblasts also upregulated two other autophagy-associated proteins, Beclin-1 and Atg5 (Figure 2A,B and Appendix A). Fibroblasts cultured in pro-autophagic condition (1% serum) were also treated with the profibrotic cytokine TGF-β. In the presence of TGF-β, cardiac fibroblasts maintained elevated levels of LC3B, Beclin-1 and Atg5 induced by the low serum medium (Figure 2A,B and Appendix A).

In order to address how autophagy regulates cardiac fibrosis, we analyzed activation of cardiac fibroblasts in the presence of BafA1, an autophagy inhibitor, which blocks fusion of autophagosome with lysosomes, leading to accumulation of autophagosomes. We confirmed that BafA1 effectively inhibited degradation of autophagosomes in cardiac fibroblasts, as indicated by accumulation of autophagosome-associated LC3B II protein in the presence of TGF-β (Appendix A). Importantly, TGF-β stimulation increased Fosl-2 protein expression (Figure 3B and Appendix A). Formation of αSMA-positive, highly contractile myofibroblasts is a hallmark of TGF-β-dependent activation of cardiac fibroblasts. We observed reduced αSMA protein levels in cardiac fibroblasts treated with BafA1 (Figure 3A and Appendix A). In line with this result, BafA1 reduced contractility of TGF-β-activated cardiac fibroblasts in the collagen gel contraction assay (Figure 3C and Appendix A). Treatment with BafA1 suppressed secretion of type I collagen in the presence and in the absence of profibrotic TGF-β (Figure 3D and Appendix A). On the molecular level, BafA1 blocked activation of the TGF-β canonical Smad-dependent pathway, as indicated by inhibited phosphorylation of Smad2 (Figure 3E). Furthermore, treatment of cardiac fibroblasts with BafA1 inhibited cell proliferation and increased caspase 3/7 activity, suggesting increased apoptosis (Figure 3F,G).

### 2.2. Fosl-2 Controls Autophagocytosis in Cardiac Fibroblasts

In the next step, we asked how the AP-1 transcription factor Fosl-2 affects autophagocytosis of cardiac fibroblasts. Analysis of cardiac tissues obtained from Fosl-2^tg^ (overexpressing *Fosl2*) mice clearly showed elevated levels of LC3B in transgenic animals (Figure 4A). Next, we isolated cardiac fibroblasts from Fosl-2^tg^ and control (Fosl-2^wt^) mice and analyzed expression of autophagy-related markers. We found that under control (10% serum) condition, Fosl-2^tg^ cardiac fibroblasts showed elevated LC3B II, Beclin-1 and Atg5 protein levels (Figure 4B). Immunofluorescent analysis confirmed increased levels of LC3B and Atg5 in Fosl-2^tg^ cells under the non-stimulatory condition and in the presence of TGF-β (Figure 4C,D). Of note, treatment with BafA1 increased LC3B II, but not autophagy adaptor protein p62 in Fosl-2^tg^ cardiac fibroblasts (Figure 4E), indicating that increased levels of LC3B II in Fosl-2^tg^ cardiac fibroblasts were caused by higher levels of ongoing autophagy rather than accumulation of autophagosomes due to impaired autophagy. Electron microscopy analysis confirmed more autophagosomes in Fosl-2^tg^ cells (Figure 4F).

Overexpression of Fosl-2 led to enhanced autophagocytosis in cardiac fibroblasts, therefore, in the next step, we analyzed the role of Fosl-2 using the loss-of-function approach. To this aim, we silenced *Fosl2* mRNA (using antisense oligonucleotides) in cardiac fibroblasts and analyzed expression of profibrotic and autophagy-related markers. The silencing successfully reduced Fosl-2 protein level by 50% (Figure 5A,B). On the one hand, *Fosl2* silencing reduced production of type I collagen at the transcript and protein levels (Figure 5C,D). Similarly, *Fosl2*-silenced fibroblasts were characterized by a lower expression of *Acta2* (gene encoding αSMA, Figure 5C), reduced αSMA total protein content (Figure 5E) and showed the absence of αSMA-positive stress fibers (Figure 5F). On the other hand, silencing of *Fosl2* suppressed autophagocytosis only under the pro-autophagic condition (1% serum) in the presence of TGF-β, as indicated by reduced protein content of Beclin-1 (Figure 5G,H). It should be noted that TGF-β is a potent activator of Fosl-2 (Figure 2B and Appendix A), which could explain the requirement of stimulatory condition to observe the Fosl-2-dependent effect in the loss-of-function approach.

### 2.3. Autophagy Regulates Fosl-2 in Cardiac Fibroblasts

As shown above, Fosl-2, under fibrotic conditions, regulated autophagocytosis, therefore, in the next step, we addressed whether blockade of autophagy process could, in turn, modulate Fosl-2 levels. Inhibition of autophagocytosis with BafA1 effectively reduced protein Fosl-2 level in cardiac fibroblasts (Figure 3B), pointing to an autoregulatory feedback loop mechanism. Next, we analyzed the role of individual autophagy proteins LC3B and Beclin-1 in regulation of Fosl-2. Silencing of LC3B in cardiac fibroblasts resulted in 50% reduced LC3B protein levels (Figure 6A). We found that cells with reduced LC3B showed less Fosl-2, however only in the presence of TGF-β (Figure 6A). In contrast, Beclin-1 levels in cells with silenced LC3B remained unaffected (Figure 6A). In our experiments, Beclin-1, similarly to LC3B, regulated Fosl-2. Silencing of Beclin-1 (50% protein reduction, Figure 6B) reduced Fosl-2 levels in cardiac fibroblasts in the presence, but not in the absence, of TGF-β (Figure 6B). Furthermore, we found that in the pro-autophagic condition, loss of Beclin-1 was associated with loss of LC3B (Figure 6B).

### 2.4. Expression of Fosl-2 and Atg5 in Cardiac Fibroblasts Control Cardiac Fibrosis in a Mouse Model of Cardiac Hypertrophy

Our in vitro data indicated that interplay between Fosl-2 and autophagy plays a crucial role in profibrotic changes in cardiac fibroblasts. To address whether these mechanisms are relevant during fibrogenesis in vivo, we analyzed fibrotic changes, using a model of experimental cardiac hypertrophy, in mice lacking either Fosl-2 or Atg5 in stromal cells using Ccl19^Cre^Fosl2^flox/flox^ and Ccl19^Cre^Atg5^flox/flox^ strains (Ccl19 is expressed in stromal cells ^25^), induced by continuous infusion of the vasoconstrictor angiotensin II delivered by osmotic minipumps. Continuous infusion of exogenous angiotensin II led to increased heart weight, collagen deposition and induced expression of Fosl-2 and Atg5 in mouse hearts (Appendix A). We observed that 3 weeks after angiotensin II infusion, hearts of Ccl19^Cre^Fosl2^flox/flox^ mice showed less collagen deposition, decreased number of gp38-expressing fibroblasts (Figure 7A,B) and reduced number of Atg5-positive cells in comparison to hearts of control Fosl2^flox/flox^ mice (Appendix A), confirming the profibrotic role of Fosl-2 in cardiac fibroblasts. A similar phenotype was found in mice lacking Atg5 in stromal cells. In the angiotensin II-mediated cardiac hypertrophy model, hearts isolated from Ccl19^Cre^Atg5^flox/flox^ mice were not only protected from increased collagen content, but also showed less gp38-expressing and Fosl-2-expressing fibroblasts and αSMA-positive myofibroblasts in comparison to controls (Figure 7C–E and Appendix A). All these results confirmed the functional involvement of Fosl-2 and autophagy in cardiac fibrosis.

## 3. Discussion

Cardiac fibrogenesis and pathological tissue remodeling represent key disease progressing processes in non-ischemic heart disease, but the exact cellular and molecular mechanisms remain elusive. Here, we demonstrated a novel mechanism, by which the TGF-β-Fosl-2 axis regulated profibrotic changes in cardiac fibroblasts by activating autophagocytosis. Previous studies have already shown the significance of autophagy in promoting fibrotic processes in multiple organs including skin [25], lung [26], liver [27], kidney [28] and heart [9,10,11]. It should be noted that in the heart, autophagocytosis might also be involved in antifibrotic and cardioprotective processes [8,12,13]. Activation of autophagocytosis is not limited to fibrotic processes but represents rather a common mechanism in many pathophysiological processes in the body. Autophagy has been also implicated in pathophysiology of inflammatory [29] and neurodegenerative diseases [30,31,32,33], cancer [34,35] and during infections [36]. Previous studies in animal models addressed the effects of systemic or cardiomyocyte-specific deregulation of autophagy in the heart [8,37,38], whereas in this work, we specifically focused on the role of autophagy in cardiac fibroblasts. We observed that fibroblast-specific deletion of the autophagic gene Atg5 effectively reduced cardiac fibrosis in a mouse model of cardiac hypertrophy. In line with our data, knock-out of autophagy activator peroxisome proliferator-activated receptor gamma coactivator-1α in fibroblasts prevented development of skin fibrosis in a mouse model of systemic sclerosis [25]. The profibrotic role of autophagy in fibroblasts was further supported by in vitro data. In particular, autophagy enhanced TGF-β-mediated conversion of cardiac fibroblasts into myofibroblast, which is in accordance with previously published results [16,17]. Similarly, in dermal fibroblasts, starvation-induced autophagy also triggered myofibroblast differentiation and enhanced production of profibrotic connective tissue growth factor [15]. Autophagy-mediated degradation of cytoplasmic components is a crucial step during cell differentiation and in phenotypic transformation [39]. This could explain the importance of autophagy in the switch of quiescent fibroblast into pathogenic myofibroblast phenotype. It should be noted that the role of autophagy in fibrosis might be more complex. It has been postulated that active autophagocytosis might prevent overproduction of misfolded collagens and other extracellular matrix proteins [40,41]. It is possible that these antifibrotic processes occur in chronic fibrosis and contribute to the fibrotic phenotype over a long period of time only. Our models, instead, reflected early fibrogenesis rather than established fibrosis. Noteworthy, we observed a functional contribution of autophagy to fibrosis in both mouse and human cells. This is not surprising, because autophagocytosis is a highly conserved evolutionary process and its involvement in common biological processes, like fibrosis, should be maintained across the species. A growing body of evidence indicates that autophagy is regulated at a transcriptional level. Transcription factor EB (TFEB), E2 transcription factor 1 (E2F1) and forkhead box O3 (FOXO) represent examples of autophagy inducers, whereas zinc finger with KRAB and SCAN domains 3 (ZKSCAN3) has been recognized as a master repressor of autophagy [42]. Furthermore, tumor protein p53 (p53) and nuclear factor kappa-light-chain-enhancer of activated B cells (NF-κB) pathways have also been implicated in regulation of autophagy [42]. In addition to known molecular regulators, our study identified Fosl-2 transcription factor as a novel direct activator of autophagy. Fosl-2 has been shown to regulate profibrotic effects of TGF-β in dermal fibroblasts of patients with systemic sclerosis and in mouse models of scleroderma [43], but its role in the regulation of autophagy has not been acknowledged until now. In line with our results, a pro-autophagic role was also demonstrated for another member of the AP-1 family, c-Jun. Published data demonstrated that c-Jun was involved in transcriptional regulation of LC3 and Beclin-1 [44], however other data indicated that c-Jun could play a repressive role in autophagy [45]. Of note, our data showed that complete blockade of autophagy with BafA1 suppressed Fosl-2 in cardiac fibroblasts, suggesting a feedback loop mechanism. In case of Beclin-1 and LC3B silencing, a downregulation of Fosl-2 was detected only under profibrotic conditions (+TGF-β). These results could be explained by the need to activate autophagy by TGF-β. Alternatively, incomplete silencing (about 50%) of these pro-autophagic proteins may cause only partial suppression of autophagocytosis. All these data encourage us to postulate that a positive feedback loop between autophagy and Fosl-2 represents an important molecular mechanism in TGF-β-mediated fibrotic response of cardiac fibroblasts. Targeting this mechanism might represent a new strategy for treatment of fibrotic heart disease.

There are, however, certain concerns regarding benefits of systemic blockade of autophagocytosis, as this process plays an important role in cardiac homeostasis. In fact, clinical data demonstrated that pharmacological autophagy inhibitors chloroquine and hydroxychloroquine could cause cardiac complications in humans [46]. However, it should be noted that chloroquine and hydroxychloroquine are not autophagy-specific drugs and activate a significant number of off-target mechanisms [47]. It seems that targeting of Fosl-2 in specific cell types, rather than autophagy, might represent a more promising strategy in the treatment of fibrotic heart disease. Inhibition of Fosl-2 could suppress not only fibrotic processes, but also autoimmune responses [22,48].

## 4. Materials and Methods

### 4.1. Animal Models

Mice overexpressing Fosl-2 (Fosl-2^tg^) were obtained from Sanofi Pharmaceuticals, CA, USA, and backcrossed with C57Bl/6 mice (Charles Rivers) for at least 10 generations. A vector containing the murine Fosl-2 gene (Exons 1 to 4, corresponding introns, and truncated untranslated regions (UTRs) under the control of the major histocompatibility complex (MHCI) promotor H2Kb was randomly inserted into the genome [22]. Wild-type (Fosl-2^wt^) littermates were used as controls. Genotype was confirmed by enhanced green fluorescent protein (EGFP) expression in blood samples. Fosl-2^fl/fl^ mice [23] were obtained from Dr. Aline Bozec (University of Erlangen, Germany) and rederivated by embryo transfer. Ccl19^Cre^ mice [49] were generated by Dr. Burkhard Ludewig (Cantonal Hospital St. Gallen, Switzerland). Ccl19^Cre^Fosl-2^fl/fl^ mice were generated by crossing Fosl-2^fl/fl^ females and Ccl19^Cre^ males for at least 2 generations in order to obtain floxed Fosl-2 on both alleles. Atg5^fl/fl^ mice [50] were kindly provided by Dr. Noboru Mizushima. Ccl19^Cre^Atg5^fl/fl^ mice were generated by crossing Atg5^fl/fl^ females and Ccl19^Cre^ males for at least 2 generations in order to obtain floxed Atg5 on both alleles. Animal experiments were performed in accordance with the Swiss federal law under the ethical approval ZH007/2010 (25.07.2019) and with the Guide for the Care and Use of Laboratory Animals, published by the US National Institutes of Health (NIH Publication, 8th Edition, 2011). Cantonal Veterinary Office Zurich approved all animal experiments.

### 4.2. Isolation and Culture of Cardiac Fibroblasts

Cardiac fibroblasts were isolated from Fosl-2^wt^ and Fosl-2^tg^ mice as previously described [18]. Briefly, mouse hearts were perfused with cold PBS, harvested mechanically and enzymatically digested by using magnetic beads in 0.025 mg/mL of Liberase (Collagenase I and II, Roche, Basel, Switzerland) solution in pure Dulbecco’s modified Eagle’s medium (DMEM) high glucose (Gibco) for 45 min at 37 °C. To inactivate Liberase activity, an equal amount of 10% fetal bovine serum (FBS, Gibco, Basel, Switzerland) in DMEM was added to the disintegrated tissues. The suspension was then filtered through a 70 µm cell strainer and centrifuged at 55 g for 2 min. Supernatants were further filtered through 40 µm cell strainers and centrifuged at 490 g for 4 min. Cell suspension was stained with anti-gp38-APC antibody (BioLegend, Clone: 8.1.1, London, United Kingdom) in fluorescence-activated cell sorting (FACS) buffer for 30 min at 4 °C, washed with FACS Buffer, centrifuged and incubated with magnetic anti-APC-microbeads (Miltenyi Biotec, Bergisch Gladbach, Deutschland) for 30 min at 4 °C. After further washing with FACS buffer, cells were sorted by an autoMACS Pro Separator (Miltenyi Biotec). Cells were plated and cultured in DMEM high glucose containing 20% FBS, 50 U/mL penicillin, 50 μg/mL streptomycin (Gibco) and 50 mM 2-mercaptoethanol (Gibco). Fetal human cardiac fibroblasts were purchased from Sigma-Aldrich, and cells of passages 10–15 were cultured as described above. In selected experiments, cells were starved substituting the culturing medium with medium containing 1% FBS, pre-treated with Bafilomycin A1 (BafA1; 50 nM, Sigma-Aldrich, Basel, Switzerland) for 4 h or stimulated with 10 ng/mL recombinant TGF-β (Peprotech, London, United Kingdom).

### 4.3. Human Endomyocardial Biopsies

Human endomyocardial biopsies (hEMBs) were obtained from patients with systemic sclerosis and inflammatory dilated cardiomyopathy (iDCM, Table 1) and patients with healed myocarditis (Table 2), which were used as controls. Samples were provided by the University Hospital Tubingen, Germany. The Ethik-Kommission der Medizinischen Fakultät des Universitätsklinikums Tubingen gave the approval to investigate endomyocardial biopsies and heart tissue which were sent for diagnostic reasons by histological and immunohistological investigations for further pathogenetic research. Approval number: 138/2004V (approval date: 9 April 2018). The experiments with re-usage of human material were approved by Swissethics (BASEC-Nr. 2019-00058, approved on 4 April 2019), and were performed in conformity with the principles outlined in the Declaration of Helsinki. The informed written consent was given prior to the inclusion of subjects in the study.

### 4.4. Gene Silencing

Cells were seeded in 6-well plates (180,000 cells/well) or 8-chamber slides (15,000 cells/well) and transfected with GapmeR or siRNA (Table 3) or the respective controls. Transient transfection was performed by 6 h incubation with Lipofectamine 2000 (Thermo Fisher Scientific) prior to treatments.

### 4.5. Contraction Assay

Fibroblast contraction in collagen gel was performed with the cell contraction assay kit (Cell Biolabs, Heidelberg, Germany) according to manufacturer’s instructions. Pictures were taken at day (d) 0, d3 and d4 using the Fusion Fx (Vilber, Eberhardzell, Germany). The areas of gels were computed with ImageJ 1.52 a.

### 4.6. Immunoblotting

Cells were trypsinized and proteins were extracted using radioimmunoprecipitation assay buffer (RIPA) Buffer (Sigma-Aldrich) complemented with protease inhibitor cocktail (cOmplete ULTRA tablets, Roche) and phosphatase inhibitors (PhosSTOP, Roche). Bicinchoninic acid assay (BCA) was used for protein quantification. 20–30 µg of proteins was used for immunoblotting analyses. Proteins were transferred on nitrocellulose membranes by wet-transfer method and incubated for 45 min in blocking solution (5% milk, Becton Dickinson, Allschwil, Switzerland) in tris-buffered saline and 0.05% Tween-20 (TBST), followed by an overnight incubation at 4 °C with primary antibodies (Table 4) and secondary antibodies conjugated with horseradish peroxidase (HRP) for 45 min. When needed, mild and harsh stripping were performed according to the protocol from Abcam.

Enhanced chemiluminescent substrate was used for the development on the Fusion Fx (Vilber). Densitometric analyses were conducted with ImageJ 1.52 a (https://imagej.net/Welcome). Fold changes were computed after normalization to glyceraldehyde 3-phosphate dehydrogenase (GAPDH).

### 4.7. Immunofluorescence Staining of Cultured Cells

Per well, 15,000 cells were seeded in glass 8-chamber slides (Milian, Boswil, Switzerland). For LC3B, Atg5 and Beclin-1 detection, cells were fixed in ice-cold 100% methanol for 10 min at −20 °C and permeabilized for 15 min with 0.1% TritonX-100 (Sigma-Aldrich). For α-smooth muscle actin (αSMA) detection, cells were fixed in ice-cold methanol:acetone in ratio 7:3 (both Sigma-Aldrich) for 10 min at −20 °C. Blocking was performed for 20 min with 10% FBS in PBS. Next, slides were incubated with the primary antibody (Table 3) diluted in PBS for 1 h at room temperature, followed by incubation with AlexaFluor 546 goat anti-rabbit IgG antibody (1:400, polyclonal, Invitrogen) or AlexaFluor 546 goat anti-mouse IgG antibody (1:400, polyclonal, Invitrogen) for 45 min at room temperature in the dark. Mounting medium (Dako, Baar, Switzerland) complemented with nuclear staining (4′,6-diamidino-2-phenylindole (DAPI 1 μg/mL, Roche)) were used to apply the cover slip to each slide. Images were acquired with an Olympus BX53 microscope equipped with a DP80 camera and edited with ImageJ 1.52 a.

### 4.8. ELISA

Pro-collagen I levels in cell culture supernatants were measured according to the manufacturer’s protocol using mouse pro-collagen I alpha 1 ELISA (Abcam, Cambridge, United Kingdom) and human pro-collagen I alpha 1 ELISA (R&D Systems, London, United Kingdom).

### 4.9. Proliferation and Apoptosis Assays

Cells were seeded in 96-well plates (20,000 cells/well): 24 h after seeding, cells were pretreated with BafA1 for 4 h, and after that, bromodeoxyuridine (BrdU, Sigma-Aldrich) was added to the culture medium for the next 24 h. Proliferation was measured as BrdU incorporation using the colorimetric cell proliferation BrdU ELISA assay (Roche) according to the manufacturer’s protocol. Apoptosis was analyzed by measuring caspase 3/7 activity using the Caspase-Glo 3/7 assay (Promega, Dubendort, Switzerland) following the manufacturer’s protocol. BrdU absorbances and Cas3/7 luminescence were measured with the Synergy HT microplate reader (BioTek, Zurich, Switzerland). Each sample was measured in quadruplicates.

### 4.10. Electron Microscopy

Cells were grown in their standard cell culture medium on 12 mm cover glasses treated with *L*-polylysine. Cells were sequentially fixed with 2.5% glutaraldehyde in 0.1 M sodium cacodylate buffer (pH 7.35, pre-warmed to 37 °C) for 1 h, with 1% OsO_4_ for 1 h in 0.1 M cacodylate buffer at 0 °C, and 2% aqueous uranyl acetate for 1 h at 4 °C. Samples were then dehydrated in an ethanol series and embedded in Epon/Araldite (Sigma-Aldrich). Ultrathin (70 nm) sections were post-stained with lead citrate and examined with a Talos 120 transmission electron microscope at an acceleration voltage of 120 KV using a Ceta digital camera and the MAPS software package (Thermo Fisher Scientific).

### 4.11. Angiotensin II-Induced Cardiac Hypertrophy Mouse Model

ALZET Osmotic minipump Model 2004 (pumping rate of 0.25 μL/h) were filled with 250 µL of 12.5 mg/mL angiotensin II (Bachem, Bubendorf, Switzerland) or NaCl and equilibrated in NaCl for 3 days at 37 °C and were implanted in 8- to 12-week-old mice. For implantation, mice received 0.05 mg/kg finadyne (Flunixin-Meglumin, Zurich, Switzerland), after which they were anesthetized by inhalation of 5% isoflurane, decreased to 1.5–2% to maintain anesthesia. During surgery, mice were placed on a heating pad and cream (Vitamin A, Baush & Lomb, Zurich, Switzerland) was applied on eyes. After shaving the upper back of the mouse and disinfecting by Betadine (Povidone-iodine), a ~1 cm incision was made to implant the minipump. The wound was closed with clips and mice were allowed to recover under a red-light lamp. Novaminsulfon (Metamizol, 200–400 mg/kg/day) was administrated in drinking water for 3 days after surgery, in order to keep pain at the minimum level.

### 4.12. Methods of Euthanasia

Mice were euthanized with 2–3 L/min influx of carbon dioxide for 5 min into the cage. After cessation of breathing, animals were left in carbon dioxide for an additional 5 min. To ensure death, animals were exsanguinated via intracardiac blood collection before tissue harvest.

### 4.13. Immunohistochemistry

Organs were fixed with 4% paraformaldehyde (PFA) for 12 h and paraffin-embedded. To detect collagen fibers, direct Red Sirius Red (Sigma-Aldrich) staining was used. Prior to immunohistochemistry, sections were boiled in citrate buffer (10 mM Citrate, 0.05% tween, pH = 6, Sigma-Aldrich) and further incubated at 95 °C for 15 min. Next, slides were incubated for 15 min in 3% H_2_O_2_ solution to block endogenous peroxidases, followed by a blocking step for one hour in 10% goat serum in Background Reducing Antibody Diluent (Dako). Avidin-Biotin Block kit (from Vector Laboratories, Servion, Switzerland) was used to block endogenous biotin. The primary antibody (Table 4) was incubated overnight at 4 °C followed by incubation with proper biotinylated secondary antibody (Vector Laboratories) for 30 min at room temperature. Sections were then incubated for 30 min with Vectastain Elite ABC solution (Vector Laboratories). Staining was developed using 3, 3 -diaminobenzidine (DAB, Vector Laboratories) followed by a counterstaining of nuclei for 1 min in Mayer’s hematoxylin solution (J.T Baker, Zurich, Switzerland). Quantification was performed on 2 consecutive cardiac sections from each mouse. The entire slide was scanned using a Slidescanner Zeiss Axio Scan microscope. ImageJ software was used to quantify positive signals.

### 4.14. Statistical Analysis

Shapiro–Wilk’s normality test was performed to evaluate the distribution of the data. Normally distributed data were analyzed by unpaired or paired two-tailed Student’s *t*-test or one-way analysis of variance (ANOVA) followed by the Fisher’s Least Significant Difference (LSD) post-hoc test, while nonparametric data were analyzed using the Mann–Whitney *U* test. All analyses were performed with the GraphPad Prism 8 software. Differences were considered statistically significant for *p* < 0.05.

## Figures and Tables

**Figure 1 ijms-22-01861-f001:**
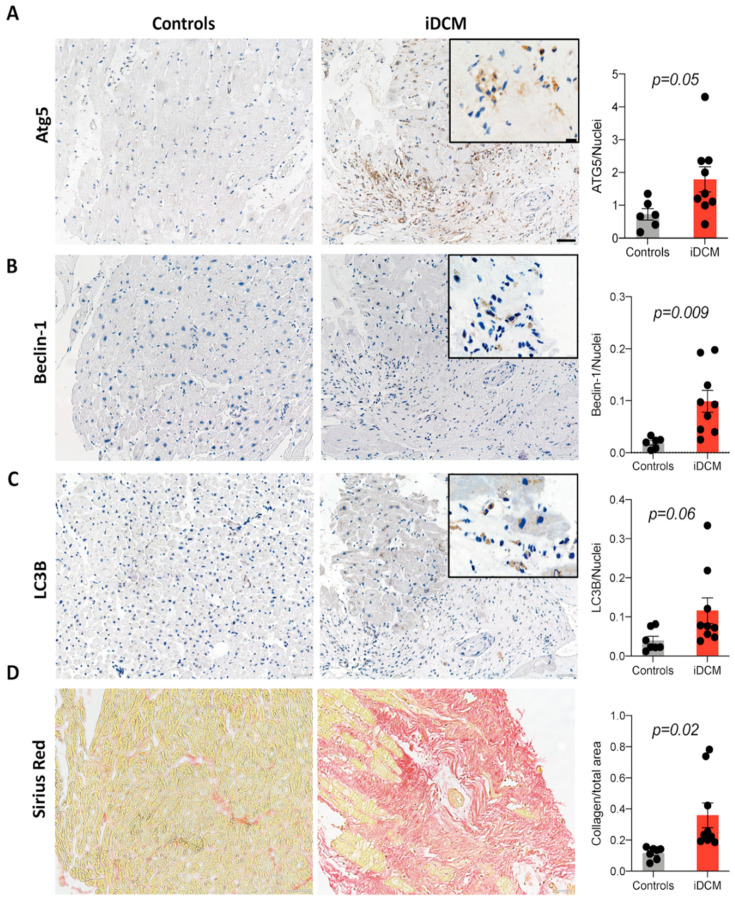
Autophagy process is activated in human hearts of iDCM patients. Representative pictures and relative quantification of ATG5 (**A**), Beclin-1 (**B**), and LC3B (**C**) IHC and Sirius Red staining (**D**) for collagen deposition in endomyocardial biopsies of controls (healed myocarditis) and iDCM patients (n = 6–9, unpaired *t*-test). Scale bars: 50 µm, scale bars in inserts: 20 µm.

**Figure 2 ijms-22-01861-f002:**
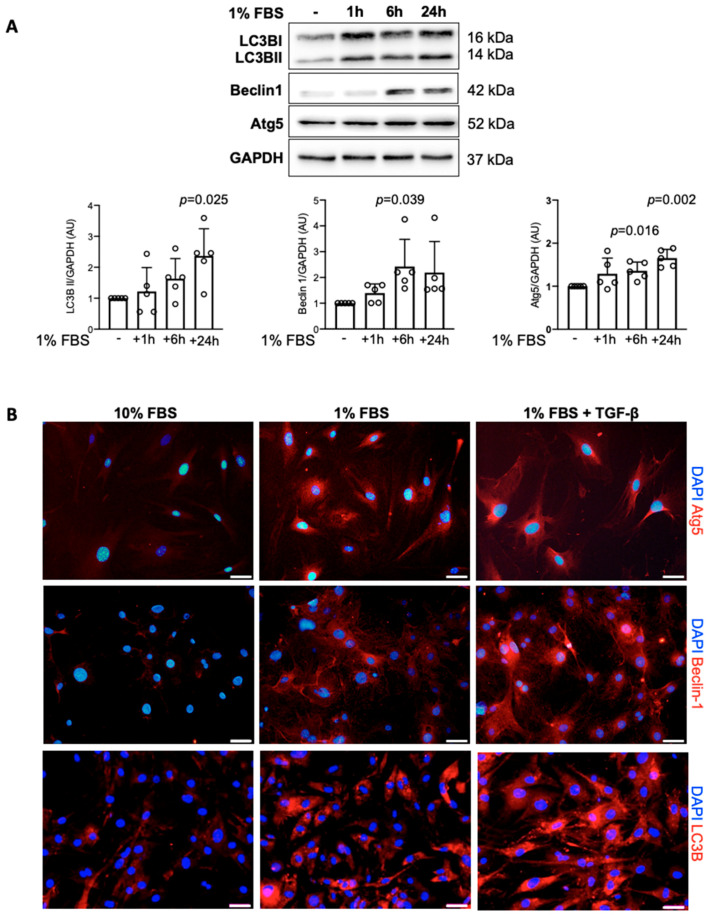
Autophagy activation in cardiac fibroblasts. (**A**) Representative pictures and densitometry analysis of immunoblots of mouse cardiac fibroblasts (one-way analysis of variance (ANOVA), n = 5). (**B**) Representative pictures of LC3B, Atg5 and Beclin-1 immunostaining (red) of cardiac fibroblasts, cultured in medium containing 10% or 1% fetal bovine serum (FBS) with or without TGF-β for 24 h (n = 3). Nuclei are stained with DAPI (blue). Scale bars: 20 µm.

**Figure 3 ijms-22-01861-f003:**
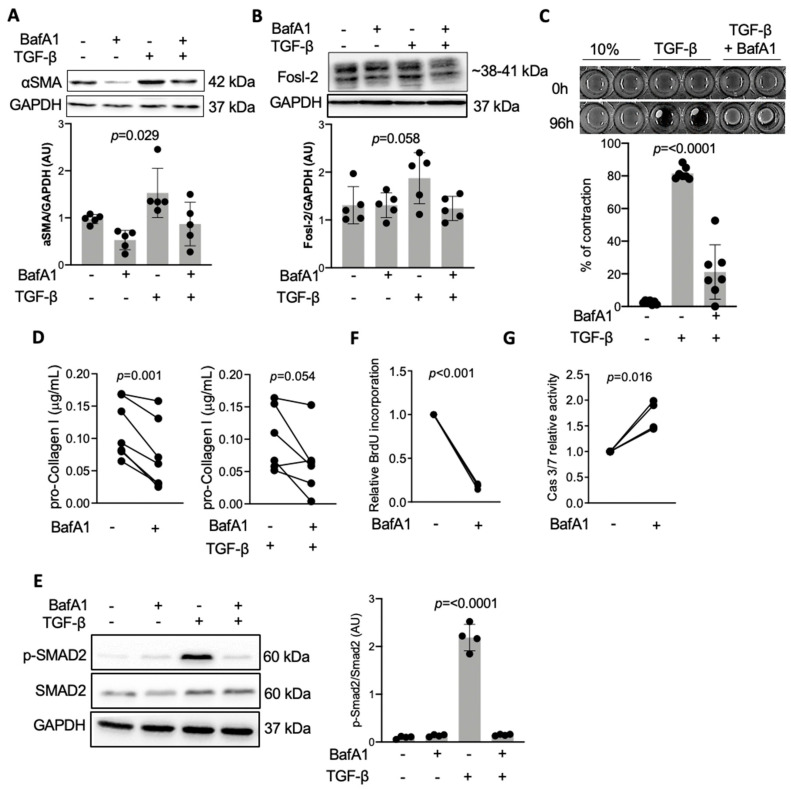
Autophagy blockage with BafA1 protects from TGF-β-induced fibroblast-to-myofibroblast transition. (**A**,**B**) Mouse cardiac fibroblasts were cultured in the presence or absence of 10 ng/mL TGF-β for 48 h with or without 4 h pre-treatment with 50 nM BafA1. Representative immunoblots and corresponding quantification of αSMA (**A**) and Fosl-2 (**B**) protein levels (one-way ANOVA, n = 5). (**C**) Representative pictures and corresponding quantification of collagen gel contraction at 0 and 96 h after seeding cells in a collagen gel. Cells were pre-treated with BafA1 for 4 h and then stimulated with TGF-β where indicated (one-way ANOVA, n = 7). (**D**) Measurements of pro-collagen I levels in supernatants of cardiac fibroblasts pre-treated with BafA1 for 4 h (paired *t*-test, n = 6). (**E**) Representative immunoblots and corresponding quantification of p-SMAD2/SMAD2 levels of cardiac fibroblasts pre-treated with BafA1 for 4 h and stimulated with TGF-β for 1 h (one-way ANOVA, n = 4). BrdU proliferation assay (**F**) and caspase 3/7 apoptosis assay (**G**) of cardiac fibroblasts pre-treated with BafA1 for 4 h (both paired *t*-test, n = 4).

**Figure 4 ijms-22-01861-f004:**
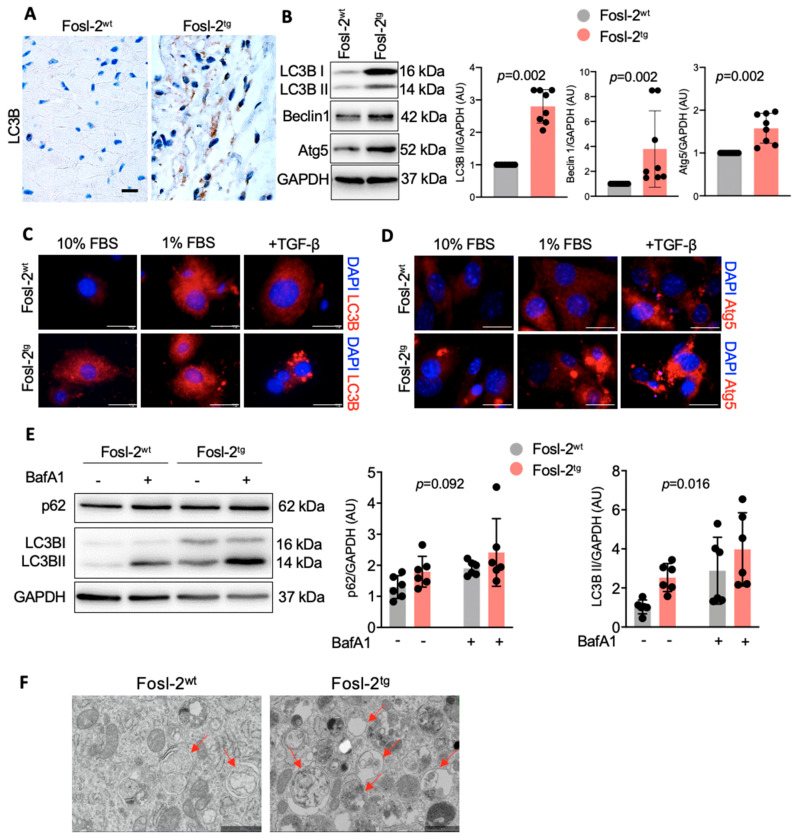
Increased autophagy in Fosl-2^tg^ cardiac fibroblasts. (**A**) Representative immunohistochemistry of LC3B in heart sections of Fosl-2^wt^ (wild-type) and Fosl-2^tg^ mice. Scale bar: 10 µm. (**B**) Representative immunoblots and corresponding densitometry of Fosl-2^wt^ and Fosl-2^tg^ cardiac fibroblasts under basal condition (Mann–Whitney *U* test, n = 8). (**C**,**D**) Representative immunofluorescence staining of LC3B (**C**, red) and Atg5 (**D**, red) of Fosl-2^wt^ and Fosl-2^tg^ cardiac fibroblasts cultured in 10% or 1% FBS with or without TGF-β stimulation for 24 h (n = 3). Nuclei are stained with DAPI (blue). Scale bars: 10 µm. (**E**) Representative immunoblots and corresponding densitometry of Fosl-2^wt^ and Fosl-2^tg^ cardiac fibroblasts treated with or without BafA1 for 4 h (one-way ANOVA, n = 7). (**F**) Electron microscopy of Fosl-2^wt^ and Fosl-2^tg^ cardiac fibroblasts treated with BafA1 for 4 h (n = 3). Arrows indicate autophagosomes. Scale bars: 1 µm.

**Figure 5 ijms-22-01861-f005:**
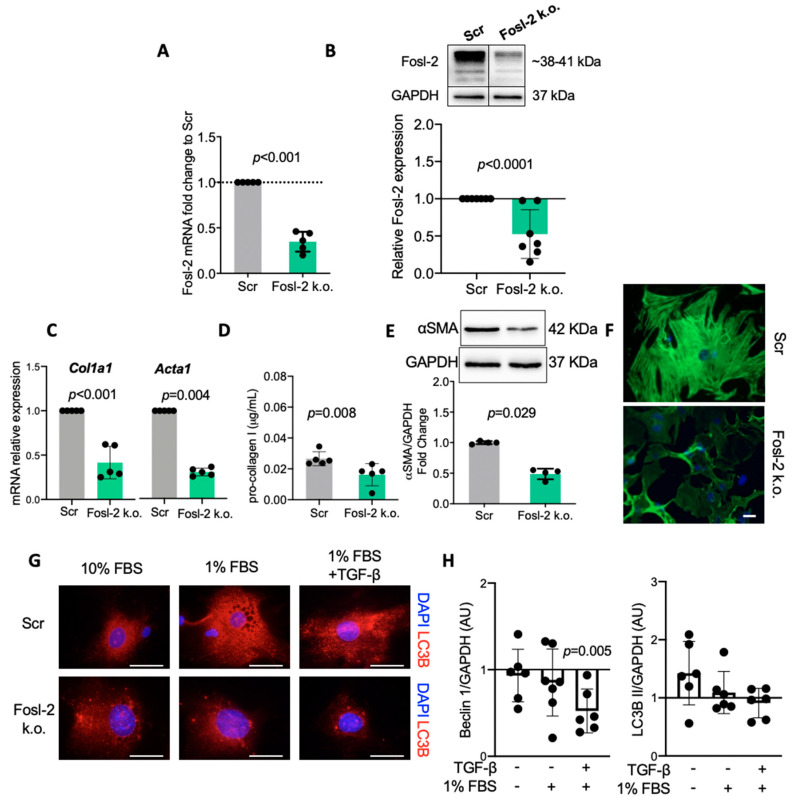
Silencing Fosl-2 suppresses autophagy in cardiac fibroblasts. Fosl-2 silencing was performed in cardiac fibroblasts using Fosl-2 antisense oligonucleotide GapmeR (Fosl-2 knockdown: k.o.) and was compared to scrambled control (Scr). Cells were cultured in starvation medium (1% FBS), unless otherwise indicated. (**A**) *Fosl2* gene expression (n = 5, Mann–Whitney *U* test), (**B**) representative immunoblots and densitometric analysis of Fosl-2 protein levels (n = 7, Mann–Whitney *U* test), (**C**) expression of genes encoding αSMA (*Acta2*) and type I collagen (*Col1a1*) compared to Scr control (n = 5, Mann–Whitney *U* test). (**D**) pro-collagen I levels in supernatants (n = 5, unpaired *t*-test), (**E**) representative immunoblots and densitometric analysis of αSMA protein levels (n = 4, Mann–Whitney *U* test). (**F**) Representative immunofluorescence staining of αSMA (n = 4, nuclei in blue are stained with DAPI, scale bars: 10 µm), (**G**) representative immunofluorescence staining of LC3B (n = 5, TGF-β treatment 24 h) and (**H**) densitometry analysis of Becn1 and LC3B II protein levels presented in relation to the respective Scr controls (n = 6, Mann–Whitney *U* test vs. Scr control).

**Figure 6 ijms-22-01861-f006:**
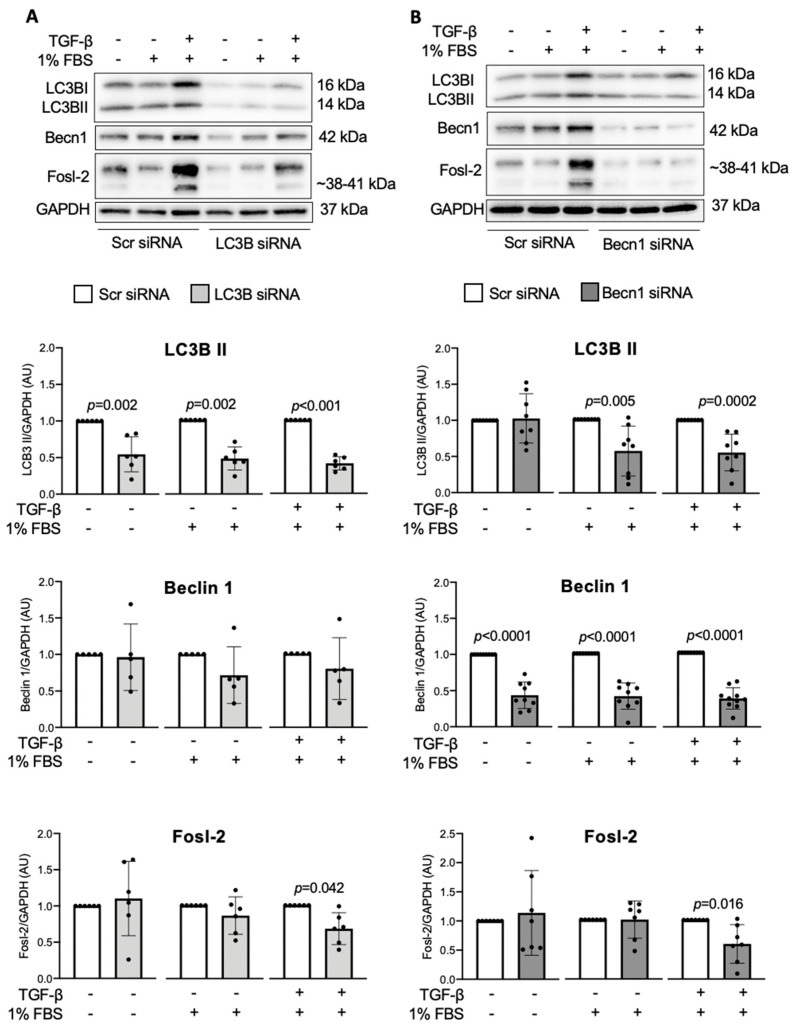
Autophagy regulates Fosl-2 in cardiac fibroblasts. LC3B and Beclin-1 silencing were performed in cardiac fibroblasts using the respective siRNAs that were compared to scrambled control (Scr). Cells were treated with medium containing 1% FBS and TGF-β for 24 h. Representative immunoblots and corresponding densitometry analysis of LC3B II, Beclin-1 and Fosl-2 protein levels following LC3B (**A**) or Beclin-1 (**B**, Becn1) silencing. Data are compared to the respective Scr control (n = 5–9, Mann–Whitney *U* test vs. Scr control).

**Figure 7 ijms-22-01861-f007:**
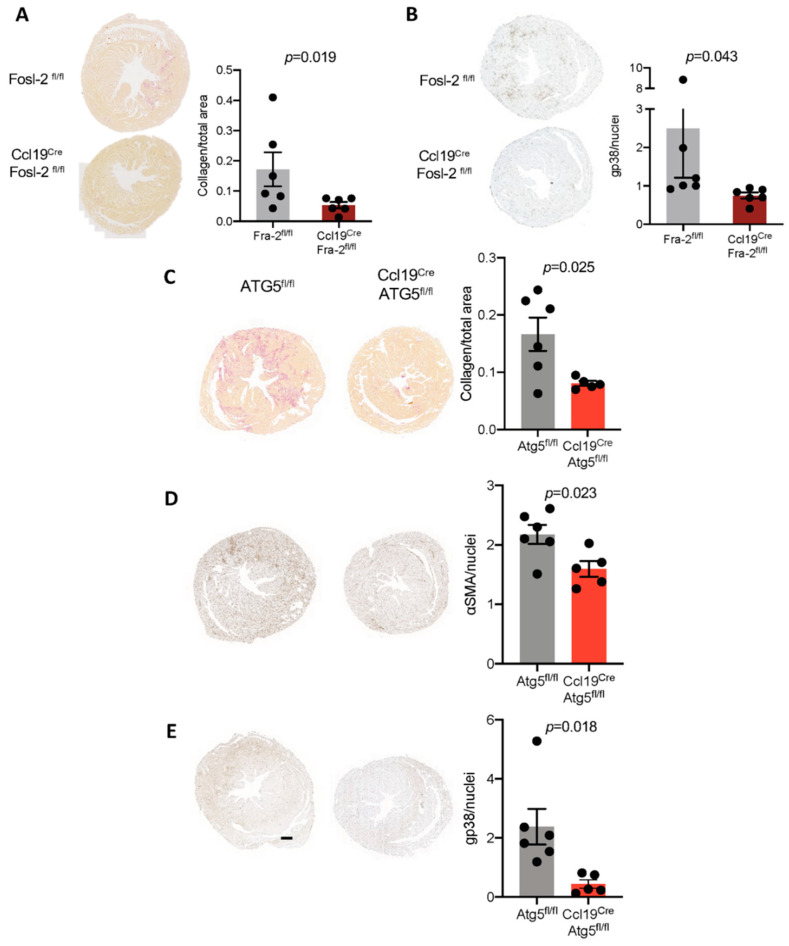
Fibroblast-specific deletion of Fosl-2 or Atg5 protects from angiotensin II-induced cardiac fibrosis. (**A**,**B**) Representative pictures and corresponding quantification of Sirius red indicating collagen deposition (**A**) and gp38 (podoplanin) immunopositive cells (**B**) of myocardial sections from Fosl-2^fl/fl^ (control) mice and Ccl19^Cre^Fosl-2^fl/fl^ mice infused with angiotensin II for 3 weeks (n = 6, Mann–Whitney *U* test). (**C**–**E**) Representative pictures and corresponding quantification of Sirius red (**C**), αSMA (**D**) and gp38 immunopositive cells (**E**) of myocardial sections from Atgf5^fl/fl^ (control) mice and Ccl19^Cre^Atg5^fl/fl^ mice infused with angiotensin II for 3 weeks (n = 5–6, Mann–Whitney *U* test). Scale bar for (**A**–**E**) (shown in **E**): 500 µm.

**Table 1 ijms-22-01861-t001:** Characteristic of iDCM patients and hEMBs.

Patient Characteristic	All EMBs (n = 9)
Age (mean ± SD years)	56.5 ± 10.5
Gender Female (n/N, %)	4/9 (44.4%)
Interstitial fibrosis (n/N, %)	9/9 (100%)
Vasculitis (n/N, %)	2/9 (22.2%)
Hypertrophy of myocytes (n/N, %)	2/9 (22.2%)
Ejection fraction (EF) < 50% (n/N, %)	4/6 (66.7%)
EF > 50% (n/N, %)	2/6 (33.3%)
HLA-DRα^+^ macrophages (n/N, %)	9/9 (100%)
CD68^+^ macrophages (n/N, %)	8/9 (88.9%)
CD3^+^ T cells (>7/mm^2^) (n/N, %)	8/9 (88.9%)
αSMA^+^ proliferating myofibroblasts (n/N, %)	6/9 (66.7%)
HLA-DRα^+^ endothelial capillaries (n/N, %)	4/9 (44.4%)
Virus negativity (n/N, %)	9/9 (100%)
Active lymphocytic myocarditis (n/N, %)	1/9 (11.1%)
Eosinophilic myocarditis (n/N, %)	0/9 (0%)
Subacute/chronic myocarditis (n/N, %)	1/9 (11.1%)
AV Block III (n/N, %)	1/9 (11.1%)

**Table 2 ijms-22-01861-t002:** Characteristic of patients with healed or not confirmed myocarditis.

Patient Characteristic	All EMBs (n = 6)
Age (mean ± SD years)	39.7 ± 19.2
Gender Female (n/N, %)	1/6 (16.7%)
Interstitial fibrosis (n/N, %)	0/6 (0%)
Active or chronic myocarditis (n/N, %)	0/6 (0%)
EF < 50% (n/N, %)	0/6 (0%)
EF > 50% (n/N, %)	6/6 (100%)

**Table 3 ijms-22-01861-t003:** Transfection reagents for gene silencing.

Target	Transfection Reagent	Conc.	Company
**Fosl-2**	Antisense longRNA (LNA) GapmeRs	150 nM	Qiagen
**Neg. control**	Antisense LNA GapmeR Control	150 nM	Qiagen
**LC3B**	ON-TARGETplus Mouse Map1lc3b siRNA	25 nM	Dharmacon
**Beclin-1**	ON-TARGETplus Mouse Beclin 1 siRNA	25 nM	Dharmacon
**Neg. control**	ON-TARGETplus Non-targeting Pool	25 nM	Dharmacon

**Table 4 ijms-22-01861-t004:** List of used antibodies.

Target	Clone	REF	Dilution	Company
**Anti-Atg5**	EPR1755(2)	ab108327	1:1000 (WB)	Abcam
**Anti-Beclin 1**	EPR20473	ab210498	1:1000 (WB)	Abcam
**Anti-LC3B**	polyclonal	2775	1:1000 (WB)	Cell Signaling
**Anti-αSMA**	1A4	A2547	1:5000 (WB)	Sigma
**Anti-Fosl-2**	REY146C	MABS1261	1:500 (WB)	EMD Millipore Corp.
**Anti-GAPDH**	14C10	2118	1:5000 (WB)	Cell Signaling
**Ter119 (PE)**	TER-119	(12-5921-81)	1:300 (FACS)	eBioscience
**CD45 (PE)**	30-F11	(12-0451-82)	1:300 (FACS)	eBioscience
**CD31 (PE)**	390	(12-0311-81)	1:300 (FACS)	eBioscience
**gp38 (APC)**	8.1.1	(127409)	1:100 (FACS)	BioLegend
**αSMA**	1A4	(A2547)	1:100 (IF)	Sigma
**αSMA**	GR3252482-2	(ab5694)	1:100 (IHC)	Abcam
**Anti-p44/p42 MAPK (ERK1/2)**	137F5	4695T	1:1000 (WB)	Cell Signaling
**Anti-p-TAK1**	S412	9339S	1:1000 (WB)	Cell Signaling
**Anti-TAK1**	polyclonal	4505S	1:1000 (WB)	Cell Signaling
**Anti-p-Smad2 (S465/467)**	138D4	3108S	1:1000 (WB)	Cell Signaling
**Anti-Smad2**	D43B4	5339S	1:1000 (WB)	Cell Signaling
**Anti-p-mTOR**	S2448	2971S	1:1000 (WB)	Cell Signaling
**Anti-mTOR**	7C10	2983S	1:1000 (WB)	Cell Signaling
**Anti-SQSTM1/p62**	GR3285981-1	ab56416	1:1000 (WB)	Abcam

## Data Availability

Not applicable.

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
