# Peer review of "The AP-1 Transcription Factor Fosl-2 Regulates Autophagy in Cardiac Fibroblasts during Myocardial Fibrogenesis"

_ijms, 2021, doi:10.3390/ijms22041861_

Round 1
Reviewer 1 Report
The AP-1 transcription factor Fosl-2 regulates autophagy in cardiac fibroblasts during myocardial fibrogenesis
Overall Review: This is an interesting study that examines autophagy in cardiac fibroblasts. While autophagy has been well-characterized in cardiomyocytes, there is less known about the role fibroblasts play in autophagocytosis during the progression of cardiac fibrosis. The authors found that the TGF-B-Fosl-2 axis regulates profibrotic changes in cardiac fibroblasts by activating autophagocytosis. Overall, the data is convincing, and the study is well-organized, detailed, thorough, and provides novel information about the potential role of fibroblasts in autophagy.
Specific comments:
- Figure 2 – the authors use * to indicate a significant p-value. However, in most of the other figures, the exact p value is listed. To be consistent, I suggest the authors change this to be consistent with the rest of the MS.
- Table 3 – Missing clone information for alpha-SMA (IHC antibody) and anti-SQSTM1/p62
- Figure 4B – Quantification shows a rather large spread of Beclin1 expression. However, for the Fosl-2WTrepresentative blot makes it appear as though it is very clear Beclin1 expression is much higher in the Fosl-2TGgroup and is therefore not very representative. I suggest using a more representative image here.
- Figure 5C – No scramble control is shown on the graph despite being mentioned in the figure legend.
- Figure 6 – It is a little confusing to determine which of the graphs belong to which blot. Headers for each graph would be nice to include. It would also be helpful if the graphs could be organized consistently for each column.
- Minor grammatical errors throughout the MS.
- Discussion section – the authors state, “these results could be explained by the need to activate autophagy and by incomplete silencing…” This sentence is confusing and needs to be rewritten for clarity. In fact, several sentences in the entire discussion section could use revision.
- Discussion section #2 – the authors write, “in this model the key role of connective tissue growth factor has been suggested.” How has the model suggested this key role of connective tissue growth factor? Further discussion is warranted here on this point.
Author Response
Responses to the Reviewer #1
We would like to thank Reviewer #1 for her/his helpful comments allowing us to improve our manuscript. Changes in the manuscript are highlighted in yellow.
Reviewer #1
Overall Review: This is an interesting study that examines autophagy in cardiac fibroblasts. While autophagy has been well-characterized in cardiomyocytes, there is less known about the role fibroblasts play in autophagocytosis during the progression of cardiac fibrosis. The authors found that the TGF-B-Fosl-2 axis regulates profibrotic changes in cardiac fibroblasts by activating autophagocytosis. Overall, the data is convincing, and the study is well-organized, detailed, thorough, and provides novel information about the potential role of fibroblasts in autophagy.
We thank Reviewer 1 for the positive feedback.
Specific comments:
- Figure 2 – the authors use * to indicate a significant p-value. However, in most of the other figures, the exact p value is listed. To be consistent, I suggest the authors change this to be consistent with the rest of the MS.
Thank you for this comment, we replaced * with the exact p values accordingly.
- Table 3 – Missing clone information for alpha-SMA (IHC antibody) and anti-SQSTM1/p62
We apologize for this mistake. The clone information has been added to Table 3.
- Figure 4B – Quantification shows a rather large spread of Beclin1 expression. However, for the Fosl-2WTrepresentative blot makes it appear as though it is very clear Beclin1 expression is much higher in the Fosl-2TGgroup and is therefore not very representative. I suggest using a more representative image here.
Thank you for this comment, we exchanged the Western blot image for the Fosl-2WT fibroblasts for the one better reflecting the statistic.
- Figure 5C – No scramble control is shown on the graph despite being mentioned in the figure legend.
The original graph represented expression of genes encoding αSMA (Acta2) and type I collagen (Col1a1) presented in relation to Scr control. Scr control was presented as a line set as 1. As such presentation of the data was confusing, we replaced the graph with more conventional one.
- Figure 6 – It is a little confusing to determine which of the graphs belong to which blot. Headers for each graph would be nice to include. It would also be helpful if the graphs could be organized consistently for each column.
We changed the type of graphs accordingly to Reviewer’s suggestion. We hope it is clearer now.
- Minor grammatical errors throughout the MS.
We apologize for these errors. English gramma will be corrected by the MDPI upon acceptance.
- Discussion section – the authors state, “these results could be explained by the need to activate autophagy and by incomplete silencing…” This sentence is confusing and needs to be rewritten for clarity. In fact, several sentences in the entire discussion section could use revision.
Thank you for this comment. We changed this sentence accordingly (lines 468-470):
“These results could be explained by the need to activate autophagy by TGF-b. Alternatively, incomplete silencing (about 50%) of these proautophagic proteins may cause only partial suppression of autophagocytosis.”
- Discussion section #2 – the authors write, “in this model the key role of connective tissue growth factor has been suggested.” How has the model suggested this key role of connective tissue growth factor? Further discussion is warranted here on this point.
We changed this sentence accordingly (lines 437-439):
“Similarly, in dermal fibroblasts starvation-induced autophagy also triggered myofibroblast differentiation and enhanced production of profibrotic connective tissue growth factor.”

Reviewer 2 Report
The data is very interesting and well described.
Major comments
It will helpful if there is a figure to show how the molecules and the function (e.g. TGF-β, autophagy, Fosl-2) relate.
There are a several minor comments.
Page 2 reference number should be superscript or written in a unified way.
Page 3 dr → Dr
Page 2-3 the number should be described in a unified way. 180'000, 15'000, 20000
Page 3 Line 233 remove the space.
Page 12 Line 391 by → in ?
Page 14 Line 452-455 The abbreviations should be fully spelled when first appear, like TFEB, E2F1, FOXO, ZKSCAN3, NF-κB.
Author Response
Responses to the Reviewer #2
We would like to thank Reviewer #1 for her/his helpful comments allowing us to improve our manuscript. Changes in the manuscript are highlighted in yellow.
Reviewer #2
The data is very interesting and well described.
We thank the Reviewer 2 for the positive feedback.
Major comments
It will helpful if there is a figure to show how the molecules and the function (e.g. TGF-β, autophagy, Fosl-2) relate.
Thank you for this suggestion. We prepared a schematic representation of molecular mechanism in the graphical abstract of the revised manuscript.
There are a several minor comments.
Page 2 reference number should be superscript or written in a unified way.
Thank you for careful reading, the reference style has been corrected.
Page 3 dr → Dr
This has been corrected
Page 2-3 the number should be described in a unified way. 180'000, 15'000, 20000
Thank you for this comment, we accordingly corrected all numbers.
Page 3 Line 233 remove the space.
It has been removed, thank you for careful reading.
Page 12 Line 391 by → in ?
Thank you for this comment, it has been corrected (in the revised manuscript in line 394)
Page 14 Line 452-455 The abbreviations should be fully spelled when first appear, like TFEB, E2F1, FOXO, ZKSCAN3, NF-κB.
Thank you for this comment, the abbreviations have been fully spelled.
